# Dynamic spin filtering at the Co/Alq$_3$ interface mediated by weakly coupled second layer molecules

Andrea Droghetti[1,2,*], Philip Thielen[3,4,*], Ivan Rungger[1,5,*], Norman Haag[3], Nicolas Großmann[3], Johannes Stöckl[3], Benjamin Stadtmüller[3,4], Martin Aeschlimann[3], Stefano Sanvito[1] & Mirko Cinchetti[3]

Spin filtering at organic-metal interfaces is often determined by the details of the interaction between the organic molecules and the inorganic magnets used as electrodes. Here we demonstrate a spin-filtering mechanism based on the dynamical spin relaxation of the long-living interface states formed by the magnet and weakly physisorbed molecules. We investigate the case of Alq$_3$ on Co and, by combining two-photon photoemission experiments with electronic structure theory, show that the observed long-time spin-dependent electron dynamics is driven by molecules in the second organic layer. The interface states formed by physisorbed molecules are not spin-split, but acquire a spin-dependent lifetime, that is the result of dynamical spin-relaxation driven by the interaction with the Co substrate. Such spin-filtering mechanism has an important role in the injection of spin-polarized carriers across the interface and their successive hopping diffusion into successive molecular layers of molecular spintronics devices.

[1] School of Physics, AMBER and CRANN Institute, Trinity College, Dublin 2, Ireland. [2] Nano-Bio Spectroscopy Group and European Theoretical Spectroscopy Facility (ETSF), Universidad del Pais Vasco CFM, CSIC-UPV/EHU-MPC & DIPC, Avenue Tolosa 72, 20018 San Sebastian, Spain. [3] Department of Physics and Research Center OPTIMAS, University of Kaiserslautern, Erwin-Schroedinger Strasse 46, 67663 Kaiserslautern, Germany. [4] Graduate School of Excellence Materials Science in Mainz, Gottlieb-Daimler-Strasse 47, 67663 Kaiserslautern, Germany. [5] National Physical Laboratory, Teddington TW11 0LW, UK. * These authors contributed equally to this work. Correspondence and requests for materials should be addressed to M.C. (email: cinchett@rhrk.uni-kl.de).

Over the last decade organic materials have emerged as a revolutionary platform for exploring spin-dependent phenomena at the nanoscale with potential for applications in data storage[1], magnetic sensing[2], and both classical[3] and quantum computing[4,5]. The study of spin physics in molecules, which has traditionally focused on spin transport in thick organic layers[6,7] and on spin manipulation at the single-molecule level[8,9], has now steadily grown into a new exciting field, called molecular spintronics[10].

One of the fundamental issues in molecular spintronics concerns the understanding of the electronic properties of hybrid interfaces formed between molecules and ferromagnetic (FM) metals[11], an aspect sometimes called the spinterface[12]. Although to date much attention has been devoted to the study of whether the spin of transition metal complexes can couple to a FM substrate[13–20], already spinterfaces comprising aromatic molecules show unique magnetic features. In fact, the hybridization between the spin-split $d$- or $f$-bands of a FM surface and the $\pi$-orbitals of a molecule leads to the formation of spin-split hybrid interface states (HISs)[21,22]. From a fundamental point of view, HISs can be seen as the fingerprint of hybridization, as they provide information about the hybridization character and strength. From a technological point of view, HISs are the determining factor for spin and charge transport across hybrid interfaces. In the simpler case of interfaces formed between organic molecules and non-magnetic materials depicted in Fig. 1a, Baldo and Forrest described charge injection in terms of a two-step process, where charges are initially injected into HISs close to the Fermi energy (first layer molecules) and successively hop from the interfacial region into the bulk distribution (second layer and beyond).

The case of interfaces formed with FM materials is sensitively more complicated, since spin-dependent hybridization will cause, in general, both a spin-dependent broadening as well as a spin-dependent energetic shift of the formed HISs. Up to now, two extreme cases have been discussed in the literature[11], depending on the strength of the hybridization between the molecules and the FM surface and the resulting energetic position of the relevant HISs with respect to the interface Fermi level ($E_F$). In the case of strong hybridization, that is, for chemisorption of the molecules in the first layer in contact to the surface, the HISs are formed close to $E_F$. Such HISs are in general spin-split (sp-HISs) and their presence changes the relative density of states (DOS) of the two spin-channels of the FM surface[23–26], as illustrated in Fig. 1b,c (first layer). This effect drastically influences spin-polarized electron tunnelling in both single-molecule[27,28] and organic spin-valve devices[29,30], and can be tailored through oxidation[31], doping[32,33] or chemical synthesis[34–38]. In contrast, in the case of weak hybridization, Fig. 1b,c (second layer), that is, for physisorbed molecules or molecules that are not in close contact with the FM surface (such as second layer molecules), the HISs are located at an energy far away from $E_F$ and, if spin-split (Fig. 1c), they form a spin-dependent potential barrier ($\varphi_\uparrow \neq \varphi_\downarrow$) that allows only one spin-channel to efficiently tunnel through the interface[39]. This gives rise to the so-called interface magneto-resistance effect, which was recently observed in devices with only one magnetic electrode[40]. Importantly, both spin-filtering mechanisms shown in Fig. 1b,c—respectively named metallic and resistive spin filtering[11]—have been discussed as completely static effects. Yet, when spin-polarized carries are injected into an organic layer and electrons are localized on HISs or on molecular orbitals for a finite amount of time, then spin and charge dynamics need to be explicitly included, namely one has to account for spin-dependent lifetimes ($\tau_{\uparrow,\downarrow}$). In a recent study, some of us[41] reported spin-dependent lifetimes of electrons

localized at the Co/Alq$_3$ interface. In particular, it was found that the majority electrons relax twice as fast as the minority ones, with $\tau_\uparrow = 450$ fs and $\tau_\downarrow = 800$ fs. In comparison, the electronic lifetimes of a clean Co surface are much shorter (some femtoseconds), and majority electrons relax slower than minority ones[42]. In ref. 41 such peculiar behaviour was attributed to the presence of a HIS 1.5 eV above $E_F$, generally called uHIS (unoccupied HIS). Crucially, the exact nature of this state and its influence on spin and charge injection across the Co/Alq$_3$ interface are still unknown.

Here we provide a combined theoretical and experimental effort to reveal the nature of the HISs at the Co/Alq$_3$ spinterface and the origin of their spin-dependent lifetimes. This study introduces the scenario depicted in Fig. 1d, where the second layer HISs are not spin-split ($\varphi_\uparrow = \varphi_\downarrow$), but possess a spin-dependent lifetime ($\tau_\uparrow \neq \tau_\downarrow$). This scenario is conceptually different from both metallic and resistive spin filtering, since the absence of spin splitting (like in Fig. 1a) will give no substantial spin-dependent tunnelling effects. However, one expects a strong influence of the HISs spin-dependent lifetime on the injection of spin-polarized carriers across the interface and their successive hopping diffusion in solid-state devices, where charge transport takes place in the organic layer[43,44]. In our investigation, we first demonstrate by density functional theory (DFT) calculations that a typical HIS at the Co/Alq$_3$ spinterface has a lifetime that is much shorter than the one reported in experiments. We, therefore, suggest that the long-living uHIS state originates from molecules weakly coupled to the surface, namely those placed above the ones in direct contact with Co (second layer molecules). Our theoretical predictions are confirmed by photoemission spectroscopy experiments on Co/Alq$_3$ samples where the second, weakly coupled, Alq$_3$ layer is desorbed making use of its lower desorption temperature as compared with the first, which is strongly hybridized. Finally, the spin-dependent dynamical properties of Co/Alq$_3$ with and without desorbed second layer are characterized by time-dependent two-photon photoemission spectroscopy. The spin-dependent lifetime of the uHIS is then explained as the result of the dynamic electron relaxation across the Co/Alq$_3$ interface driven by the interaction of the molecules with the Co substrate. We predict that this dynamic effect will have the dominant role in injection-based devices, where electrons hop through the Co/Alq$_3$ interface. Our results, together with those recently published by Raman et al.[40] demonstrate that, while much research effort has been concentrated on strengthening the chemisorption process for molecules on FM substrates[35,36], the design of weakly coupled organic layers to be deposited on top of spinterfaces may represent a different strategy for tuning the spin-dependent transport characteristics of molecular spintronic devices in both the tunnelling as well as the injection/hopping regimes.

## Results

**Theoretical calculations.** We start by computing the typical lifetime, $\tau$, of a HIS originating from the hybridization of the Alq$_3$ lowest unoccupied molecular orbital (LUMO) with the Co surface. This is obtained as $\tau = \hbar/\Gamma$ from the broadening, $\Gamma$, of the Alq$_3$ DOS induced by coupling the molecule to a semi-infinite Co substrate (see calculations details in the Methods section).

The most energetically favorable adsorption geometry for an Alq$_3$ molecule on Co is shown in Fig. 2b, where two quinoline ligands lie almost flat on the surface, while the third one points away from it[37]. For this configuration two of the oxygen and one of the nitrogen atoms coordinating the central aluminum form a strong covalent bond with the surface, while the other nitrogen

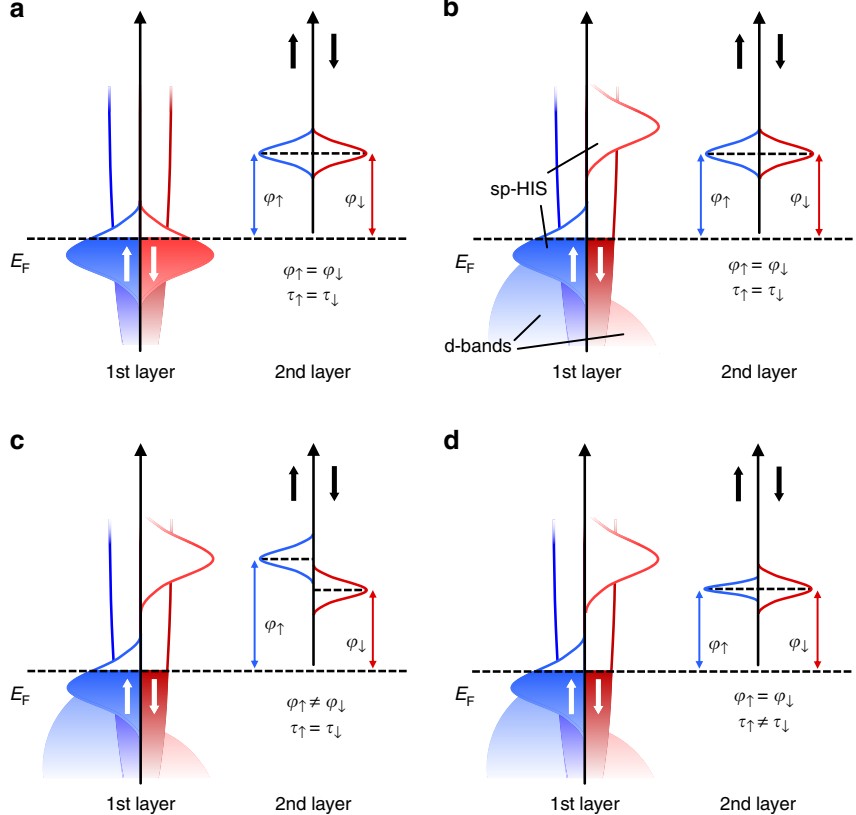

**Figure 1 | Spin and charge transport mechanisms across hybrid interfaces.** When an organic material is deposited on a metallic substrate, the large hybridization between molecules of the first organic layer and the substrate leads to the formation of a broad HIS at the interface Fermi energy. In contrast, for molecules in the second layer, that have a much weaker hybridization with the substrate, the HIS has a much more pronuced molecular character and is located at an energy far away from the Fermi level. Then the difference between the energy of the HIS and the Fermi energy defines the potential barrier φ that must be overcome to achieve charge injection (**a**). If the substrate is ferromagnetic, different mechanisms for spin-transport through the interface are possibile. In case of tunnelling transport, metallic spin filtering (**b**) is caused by the spin-split (sp) HIS stemming from the first organic layer that is located at the interface Fermi level. In contrast, resistive spin filtering (**c**) is due to tunnelling of electrons across the spin-dependent potential barrier ($\varphi_\uparrow \neq \varphi_\downarrow$) due to the energy difference between the spin-split second layer HIS and the Fermi level. In case of spin-injection and not tunnelling transport, dynamic spin filtering (**d**) occurs when electrons are injected into the HIS of the molecules in the second layer. This HIS is not necessarily spin-split, but presents a spin-dependent lifetime.

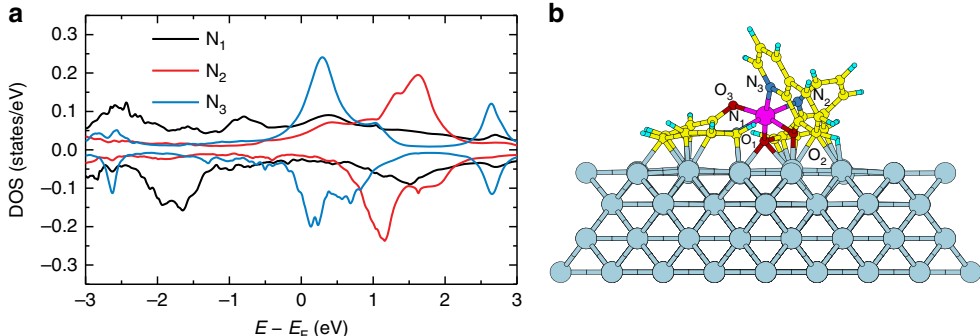

**Figure 2 | DFT calculations for the hybridization of the Alq$_3$ LUMO with the Co surface.** (**a**) Density of states projected over the Alq$_3$ 2p-N$_1$, N$_2$, N$_3$ orbitals, which give the dominant contribution to three Alq$_3$ LUMOs. The LUMO originating from the 2p-N$_1$ orbitals is broadened into a band extending over the whole plotted energy range (black DOS). The other two LUMOs, which present a large contribution from the 2p orbitals of N$_2$ and N$_3$ (red and blue DOS), appear as two Lorentzian-like peaks centered at $E - E_F \approx 1.4$ eV and $E - E_F \approx 0.3$ eV, respectively. The energy splitting between these two peaks is largely affected by the distortion of the ligands, which determines a shift of the LUMO localized over N$_2$ towards higher energies than that of the LUMO localized over N$_3$. (**b**) DFT-optimized geometry for the Co/Alq$_3$ interface. The two oxygen atoms and the nitrogen atom forming a strong covalent bond with the surface are labelled O$_1$, O$_2$ and N$_1$. In contrast, the atoms labelled N$_2$, N$_3$ and O$_3$ are much more separated from the surface. Colour code for the Co/Alq$_3$ interface: Co (grey), Al (purple), C (yellow), O (red), N (dark blue) and H (cyan).

and oxygen atoms are much more separated. In the gas phase, the three LUMOs of $Alq_3$ are almost degenerate and localized mainly on the pyridyl moiety of each ligand, so that their dominant contribution originates from the $2p$ orbitals of each nitrogen[45]. The adsorption on Co leads to different splitting and broadening of the three LUMOs (Fig. 2a) depending on the strength of the Co-ligand coupling and on the distortion of the ligands. As described elsewhere[37], a similar effect occurs also for the three highest occupied molecular orbitals (HOMOs), which are localized on the phenoxyl moiety of each ligand, with dominant contribution from the oxygen atoms. From Fig. 2a, it is immediately evident that the substrate-induced broadening $\Gamma$ is very large. Indeed, the computed value for the three LUMOs is between 0.5 and 1 eV, which translates into lifetimes ranging only from 0.6 to 1.2 fs. This estimate is two orders of magnitude smaller than the value reported in experiments[41], and compares with the typically measured lifetime of metallic surfaces[46]. We, therefore, conclude that the strongly chemisorbed $Alq_3$ molecule becomes effectively a component of the metallic hybrid surface, altering the spin-dependent DOS around $E_F$, but not acting as a spin-dependent electron trap. States with a long lifetime, therefore, must be associated to molecules, which have a much weaker coupling to the surface.

A further insight into the nature of the uHIS state measured in two-photon photoemission (2PPE) is obtained by noting that in experiments the molecules composing a nominal $Alq_3$ single layer (one monolayer, ML) do not grow ordered on the Co surface, mainly due to the three-dimensional geometric structure of $Alq_3$ (ref. 47). The theoretical results suggest that the experimental one ML film comprises chemisorbed molecules, as well as molecules, which are only physisorbed, and which potentially fill the empty spaces on the surface and thus maximize the packing density. In this context, we define the strongly hybridized molecules in direct contact with the surface as the real first molecular layer, and the weakly adsorbed molecules as the second molecular layer. While the DOS of the first layer presents broad HISs (Fig. 2a), the DOS of the second layer can be expected to display much more distinct molecular features and consequently much longer lifetimes. Then the question becomes how long these lifetimes are and whether they are of the same order of magnitude of those measured by 2PPE.

In order to verify this hypothesis, we construct a model system, which accounts for both the first and the second layers defined above, and we then compute the energy-position and lifetime of the LUMO of the molecules in the second layer. This model consists of a set of fully optimized supercells with two embedded molecules: one strongly adsorbed on the surface and the other slightly more detached, so not to form a covalent bond with Co. In Fig. 3 one such supercell is displayed. The binding energy per atom for the second layer molecule is found to be 30 meV, while for the molecules in the first layer it is one order of magnitude larger, which indicates that the second layer molecules will desorb significantly easier than the ones of the first layer. The energy-position of the LUMO and the DOS are computed with a well-established approach, which is usually employed for electronic transport simulations in molecular junctions. First, the correct position of the LUMO, which also includes the shift induced by image charge effects[48,49], is inferred from the total energies of constrained-DFT (c-DFT) calculations[50,51], and then the DFT Kohn–Sham eigenvalues are shifted to the correct energy-positions through the application of a scissor operator[51,52] (see the Methods section for a detailed description of the method and a discussion of its appropriateness when dealing with unoccupied molecular states).

The results are summarized in Fig. 4a, which shows the theoretically predicted energy level alignment diagram of the Co/$Alq_3$ interface. This diagram indicates the change in the Co

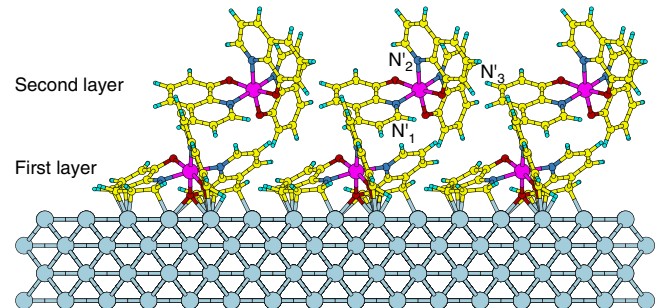

**Figure 3 | Schematic representation of the arrangement of the $Alq_3$ molecules on the Co surface.** For a nominal coverage of one ML, we distinguish between molecules in direct contact with the Co surface (first layer) and molecules that are far away from Co and thus are weakly coupled (second layer). Here we display the supercell describing the first and second layer of the Co/$Alq_3$ interface (periodically repeated three times along one direction for better display). The nitrogen atoms in the ligands of the physisorbed $Alq_3$ molecule are labelled $N'_1$, $N'_2$ and $N'_3$ for future reference. Colour code for the Co/$Alq_3$ interface: Co (grey), Al (purple), C (yellow), O (red), N (dark blue) and H (cyan).

work function due to the formation of the interface dipole $(\Delta)$[37], the broad band of the HISs associated to the chemisorbed molecules (first layer), the second—layer HOMO computed with c-DFT, and the $Alq_3$ HOMO–LUMO gap in the gas phase[37,53]. Note that, in order to account for the amorphous character of the $Alq_3$ layers, we considered ten different fully optimized supercells, where the two molecules have different relative positions, and the reported results represent the average over these supercells.

When the calculated results are compared with the experimental energy level alignment diagram as obtained in previous works[37,41] and depicted in Fig. 4b, the most striking outcome is that the calculated average energy of the second layer LUMO is $E - E_F \approx 1.5$ eV, which is approximately equal to the energy of the long-living unoccupied uHIS state measured by 2PPE[41]. This is a first verification of the hypothesis that the long-living state measured in 2PPE experiments is indeed the second-layer LUMO of the Co/$Alq_3$ interface, and that the measured large energetic broadening[41] is caused by the relative shift of individual LUMO energies in the varying local environment of the highly disordered second layer (inhomogeneous broadening). Further support to our interpretation comes from the long lifetimes of the $Alq_3$ second-layer. The lifetime of these states is evaluated from the broadening $\Gamma$ of the DOS after applying the scissor operator to move the DFT eigenvalues to the energies obtained from c-DFT. The DOS of the three LUMOs, shown in Fig. 5 for the representative supercell displayed in Fig. 3, presents an average $\Gamma$ of $\sim 3$ to 6 meV, which implies a $\tau$ of 100–300 fs, in good agreement with the experimental values. We note that the computed $\tau$ is similar to that we would have obtained without shifting the LUMOs, since these states are mainly coupled to the broad Co $s$ band, which is spread over a very wide energy region. To verify the robustness of the results we have repeated the calculation for an increased (decreased) second layer molecule-substrate distance by 0.5 Å, and we have found that the changes in the computed broadening are within a factor two of the ones calculated at the relaxed distance, so that overall the predicted order of magnitude for the lifetime does not change. This demonstrates that our results are robust against systematic errors deriving from the employed computational approach for the structural relaxation.

To summarize, the overall picture that emerges from our theoretical analysis is the following: one nominal ML of $Alq_3$ is composed of both chemisorbed (first layer) and physisorbed

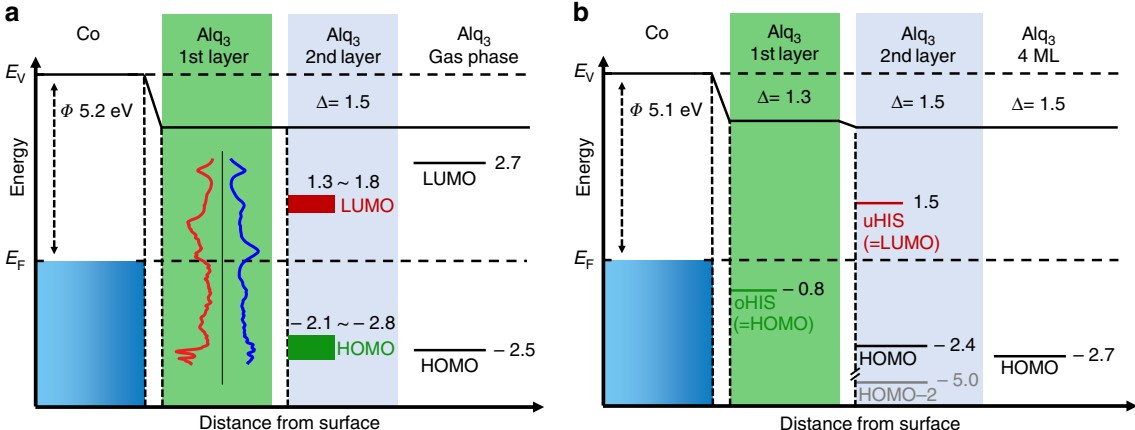

**Figure 4 | Energy level alignment at the Co/Alq₃ interface. (a)** Theory, **(b)** experiment. The experimental results have been taken from the authors' previous works (refs 37,41 Note in particular the presence of the uHIS state probed by 2PPE at $E - E_F \approx 1.5$ eV: the second layer LUMO. (The LUMO is not detected by 2PPE for a nominal coverage of Alq₃ above four ML.) In the theoretical diagram, the work function is evaluated with DFT as the difference between the Fermi energy and the reference Hartree potential in the vacuum region far away from the surface. The total density of states of the strongly chemisorbed first layer is computed with DFT for the optimized geometry of the Co/Alq₃ interface. The second layer HOMO and LUMO energy-position are calculated from c-DFT. The gas phase HOMO–LUMO gap is obtained through finite differences with the ΔSCF (self-consistent field) method[32].

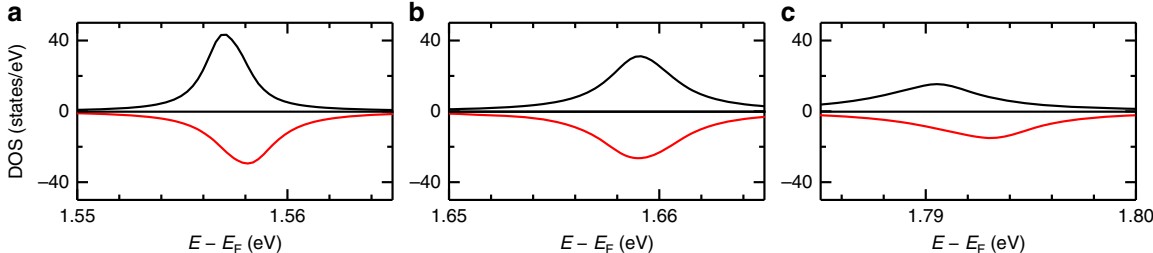

**Figure 5 | DOS for the three LUMOs of a representative Alq₃ molecule in the second layer.** Panel **(a)** corresponds to the LUMO with a large contribution from N′₁, **(b)** with a large contribution from N′₂ and **(c)** with a large contribution from N′₃.

(second layer) molecules. While the first layer forms a background of broad HISs that change the metal DOS over a wide energy range around $E_F$, the second layer shows sharp molecular features with associated very long lifetimes.

**Desorption experiments.** Experimental verification of our hypothesis requires distinguishing between chemisorbed and physisorbed molecules. A freshly prepared nominal ML of Alq₃ will contain molecules bound either way as Alq₃ grows disordered on Co[47]. The physisorbed molecules can then be removed by carefully heating the sample, since their bond to the substrate is much weaker than that of the chemisorbed molecules and thus desorb at lower temperatures. In order to avoid atom interdiffusion from the substrate to the Co/Alq₃ interface during heating, we use here an Au(111) crystal as substrate for the FM Co film instead of Cu(001) used in previous works[31,35,37,41].

Desorption of the physisorbed molecules was achieved in consecutive heating cycles: the sample was heated stepwise for ten minutes to 160/180/200/220/245 °C. In between each step the evolution of the molecular features close to $E_F$ was monitored with ultraviolet photoelectron spectroscopy (UPS, see Methods). The UPS results are shown in Fig. 6a. In the as-grown (one nominal ML) Alq₃/Co sample, the most dominant molecular features are the peaks at $E - E_F = -2.4$ eV and $E - E_F = -5.0$ eV. For increasing heating temperature, the spectral intensity of these features decreases. The peaks almost disappear at the highest annealing temperature ($T = 245$ °C). We can thus ascribe those

peaks to the HOMO and HOMO-2 of the second-layer Alq₃ molecules physisorbed on Co, in agreement with the findings in ref. 37. Note that the UPS spectra of the annealed samples still differ from the bare Co spectrum, indicating that the strong chemisorbed first-layer Alq₃ molecules are still present on the Co surface.

In order to investigate the unoccupied states, lying energetically between the Fermi level and the vacuum level of the system, we used 2PPE (see Methods). Fig. 6b shows the 2PPE spectra of the as-grown Co/Alq₃ sample compared with the sample heated at 245 °C. The broad spectral feature around $E - E_F = 1.5$ eV, corresponding to the uHIS state discussed in ref. 41, is strongly quenched in the 2PPE spectra of the heated sample, showing thus the same behaviour as the spectral features observed with UPS. This means that the peaks at –2.4 eV and –5.0 eV in the UPS spectra, as well as the peak at 1.5 eV in the 2PPE spectrum originate from physisorbed second-layer Alq₃ molecules. From the position of the low-energy cutoff of the 2PPE spectra—which is a measure of the samples work function—we also observe that the work function of the heated Co/Alq₃ sample is 1 eV lower than the work function of the bare Co surface, giving the second indication that the chemisorbed first layer Alq₃ molecules are still on the surface of the heated sample.

We now concentrate on the spectral feature at around $E - E_F = 1.5$ eV in the 2PPE spectra. Although we have demonstrated that this feature is related to second layer Alq₃ molecules, we still have to prove that the long lifetimes reported in ref. 41 are related to this second layer uHIS. To this end, we

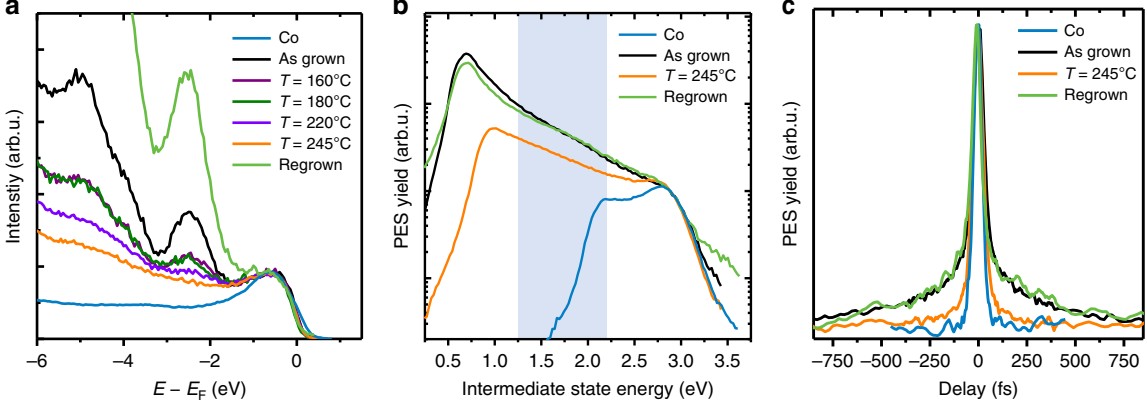

**Figure 6 | Disentangling the spectroscopic contribution of first- and second-layer Alq₃ molecules.** (**a**) Comparison between the UPS spectra of the bare cobalt substrate with the UPS spectra of the Co/Alq₃ sample (as-grown, heated at temperatures between 160 °C and 245 °C, and re-grown). The UPS spectra of the as grown sample contain molecular features of both physisorbed (second-layer) and chemisorbed (first-layer) Alq₃ molecules. After heating at 245 °C most of the physisorbed molecules are desorbed. (**b**) Comparison between the static 2PPE spectra of the bare Co substrate with the spectra of the Co/Alq₃ sample (as-grown, heated at 245 °C, and re-grown). The energetic position of the uHIS is highlighted by the light-blue area. (**c**) Time-resolved 2PPE autocorrelation traces recorded at the energetic position of the uHIS from the bare Co substrate, and from the Co/Alq₃ sample (as-grown, heated at 245 °C and re-grown). arb. u. arbitrary units.

have performed time-resolved 2PPE experiments (see Methods) on the bare Co, the as-grown Co/Alq₃ sample, and the Co/Alq₃ sample heated at 245 °C. The results are reported in Fig. 6c. Crucially, the 2PPE autocorrelation traces of the as-grown and annealed samples are drastically different. In particular, the autocorrelation trace of the annealed sample shows a much faster decay in the wings of the 2PPE signal at delays > 200 fs, which are the fingerprints of the long-living uHIS[41]. Thus, the time-resolved 2PPE experiments confirm our hypothesis: the uHIS at around $E - E_F = 1.5$ eV and its long lifetime are indeed related to weakly bound physisorbed Alq₃ molecules, that is, to molecules from the second layer.

To further confirm our conclusions, we have re-evaporated 0.5 ML of Alq₃ on the Co/Alq₃ sample after the last heating step at 245 °C. After re-evaporation of Alq₃ we can recover all the spectral features related to second-layer Alq₃ molecules: (i) the peaks in the UPS spectra at −2.4 eV and –5.0 eV (Fig. 6a); (ii) the uHIS feature in the 2PPE spectra (Fig. 6b); (iii) and the autocorrelation wings related to the long uHIS lifetime (Fig. 6c). All these observations confirm that heating the as-grown Co/Alq₃ sample to $T = 245$ °C leads to the desorption of physisorbed Alq₃ molecules, that is, of the second layer. The disappearance of both the uHIS state in the 2PPE spectrum as well as of the fingerprint of its long lifetime in the autocorrelation trace of the heated sample confirm that the long-living uHIS state can be ascribed to the physisorbed second-layer molecules, in full agreement with our theoretical calculations (see the comparison between theory and experiment in Fig. 4). According to theory, the second-layer uHIS can be thus identified with the LUMO of the physisorbed second-layer Alq₃ molecules at the Co/Alq₃ interface.

## Discussion

So far we have mainly addressed the origin of the long lifetime reported for the uHIS of the Co/Alq₃ interface, demonstrating that it is related to the physisorbed second layer of Alq₃ molecules. However, we have not discussed its spin-dependent properties, which were presented in ref. 41, and which indicate that minority electrons have a lifetime twice as long as that of the majority ones: $\tau_{majority} = 450$ fs, $\tau_{minority} = 800$ fs (majority and minority components are defined with respect to the Co DOS). This quantitative observation cannot be understood from the interface DOS only.

In Fig. 5 we see that, while the molecular states are slightly spin-split, $\Gamma$ is hardly spin-dependent, and in fact for the minority spins it is slightly larger than for the majority ones, implying a slightly shorter lifetime (this is opposite to what is observed experimentally). This difference is due to the fact that in the Co DOS the minority d-bands extend up to $E - E_F \approx 1.5$ eV. Therefore, a slightly larger hybridization, leading to a larger $\Gamma$, is found for minority unoccupied molecular states than for majority ones (see Fig. 7a,b). Besides that, we note that the larger broadening of the minority molecular states with respect to the majority ones was predicted also by Raman et al.[11] for the second layer of DMP molecules on a Co surface. However, in their calculations, the predicted splitting between the majority and minority states is of the order of 100 meV, a factor ten larger than in the present case (compare Fig. 1b with Fig. 5). We thus argue that the smaller splitting of the Co/Alq₃ uHIS prevents a resistive-spin-filtering behaviour in devices where electrons tunnel through the Co/Alq₃ interface.

All these results indicate that the measured lifetime difference cannot be simply explained by the broadening of the Alq₃ molecular orbitals due to the hybridization with the Co substrate. The origin of the measured spin-dependent lifetime must thus be a dynamical effect related to the spin-dependent relaxation of electrons residing on second-layer molecules of the Co/Alq₃-interface.

Figure 7 shows a schematic representation of the possible microscopic energy-and spin-relaxation channels at the Co/Alq₃ interface. At the time $t = 0$, the pump laser pulse in the 2PPE experiments creates a transient positive spin polarization in the uHIS, that is, a spin ensemble with an unbalanced majority spin population. After the excitation, that is, for $t > 0$, this ensemble starts to relax via two main processes[54]: (i) elastic and/or inelastic scattering into the Co surface or the first-layer Alq₃ molecules; (ii) quasi-elastic spin-flip scattering. The first process (i) leads to a depopulation of the uHIS. The second process (ii) does not change the population of the uHIS, but instead alters its spin polarization. If we assume that the scattering associated with the two processes is spin-independent, we can model the temporal behaviour of the population of the majority and minority spin channels with two coupled differential equations, containing two parameters: the depopulation time $\tau_{dep}$ and the spin-flip time $\tau_{sf}$ (see Methods). In Fig. 7c we plot the transient population of the majority and minority channel of the uHIS obtained by solving the differential equations for $\tau_{dep} = 600$ fs and $\tau_{sf} = 1000$ fs

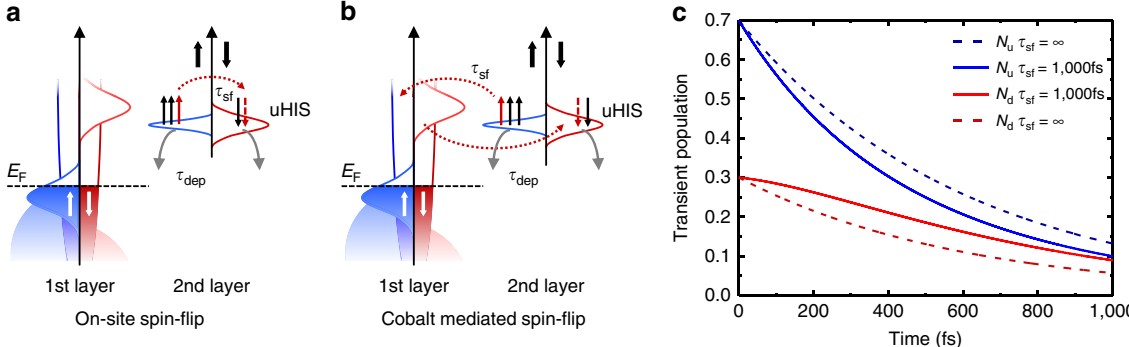

**Figure 7 | Schematic representation of dynamical spin filtering at the Co/Alq$_3$ uHIS.** The panels (**a,b**) show schematically the processes leading to depopulation ($\tau_{dep}$) and depolarization ($\tau_{sf}$) of the uHIS. There are two possible spin-flip mechanisms leading to depolarization of the uHIS: (**a**) on-site quasi-elastic spin-flip scattering, mediated by either spin-orbit coupling or hyperfine interaction on the uHIS; or (**b**) quasi-elastic spin-flip scattering mediated by the Co substrate. The panel (**c**) shows the transient population of the uHIS simulated by assuming an initial spin polarization of 40% and setting $\tau_{dep} = 600$ fs and $\tau_{sf} = \infty$ (dashed lines) or $\tau_{sf} = 1000$ fs (continuous lines). For a finite spin-flip scattering rate, the population of the minority spin channel is transiently increased in the time range between 200 and 800 fs due to refilling of minority electrons that results from the activated spin-flip channel.

(continuous lines). These values are chosen by taking $\tau_{dep}$ as the average uHIS lifetime measured in ref. 41 and by assuming that the quasi-elastic spin-flip time is longer than the depopulation rate, in line with ref. 54. The solution of the rate equations are plotted in Fig. 7, together with the single exponential decay of the population that would result from $\tau_{sf} = \infty$, that is, in absence of spin-flip scattering (dashed lines). From the figure it is clear that, for an initial population containing more majority electrons, quasi-elastic spin-flip processes cause a 'refilling' of minority electrons from the majority spin channel[55]. Such a refilling process leads to an effective slower decay rate for the minority spin channel than for the majority channel for time delays shorter than $\tau_{sf}$. We can thus conclude that quasi-elastic spin-flip scattering is the origin of the spin-dependent lifetimes measured in ref. 41. Although spin-independent scattering constants can explain the results at the Co/Alq$_3$ interface, we would like to point out that in general $\tau_{sf}$ does not need to be spin-independent. A spin-dependent $\tau_{sf}$ can cause an even stronger 'refilling' either in the majority or minority spin channel, thus leading to a transient enhancement or even inversion of the uHIS original spin polarization.

To summarize, HISs formed with second layer Alq$_3$ molecules that are physisorbed on the cobalt substrate, acquire a spin-dependent lifetime as the result of dynamical spin-relaxation driven by the interaction with the substrate: after initial population with an ensemble of positively spin-polarized electrons, the minority channel is re-filled by majority electrons that have flipped their spin after interaction with Co. This effect slows down the depopulation rate of the minority channel, and thus strongly influences the hopping diffusion and the injection of spin-polarized carriers across the interface. For this reason, we name this effect 'dynamic spin filtering'. Dynamic spin filtering likely affects spintronic devices and we envisage future studies aimed at addressing its impact in transport experiments.

The final question still open is about the microscopic processes leading to quasi-elastic spin-flip in the second-layer uHIS. As schematically depicted in Fig. 7, we distinguish between two possibilities: (a) on-site and (b) off-sites scattering. On-site elastic spin-flip processes, that is, scattering processes taking place on the uHIS molecular level, could be either spin-orbit- or hyperfine interaction-assisted scattering. In contrast, off-site scattering could be given by vibron-assisted spin-flips as described recently in ref. 56. The latter process does not need any source of spin-depolarization at the molecular site, as the spin-flip processes are mediated by the Co substrate.

Although the exact determination of which microscopic mechanism leads to a more effective refilling of the uHIS is beyond the scope of this paper, our results uncover a dynamic spin-filtering effect that may have a dominant role in injection-based devices where electrons hop through hybrid interfaces. We predict that choosing specifically designed molecular components as weakly coupled second layer, such as molecules with a large intrinsic magnetic moment or molecules magnetically coupled to the first layer[57], will induce strong spin-dependent quasi-elastic spin-flip scattering, and thus strongly influence the size and sign of the spin polarization injected across the interface. In general, physisorbed second-layer molecules constitute a very promising playground for the design of advanced molecular spintronics concepts. First of all, they mostly keep their molecular character, contrary to the case of chemisorbed molecules; second, their physical properties can be potentially controlled by external stimuli (such as light, doping, electric or magnetic fields), offering the opportunity to control the related interfacial spin-filtering properties. Our findings thus indicate that, while much research effort has been concentrated on strengthening the chemisorption process for molecules on FM substrates[35,36], the design of weakly coupled organic layers to be deposited on top of spinterfaces may represent a strategy for tuning the spin-dependent transport characteristics of molecular spintronic devices.

## Methods

**First-principles calculations.** The DOS of the Co/Alq$_3$ interface is computed by using the Smeagol code[58,59], which combines DFT[60] with the non-equilibrium Green functions (NEGF) method for electron transport[61], often referred to as DFT + NEGF. Within this framework, the Co substrate can be modelled as a semi-infinite electrode rather than as a finite slab. Mathematically, the coupling between the molecules and this electrode is described by the so-called leads' self-energy, which acts as an energy- and spin-dependent non-Hermitian potential[58,59], allowing for in- and out-flow of electrons from/to the Alq$_3$ to/from the Co bulk. To a first approximation the DOS, $N_n(E)$, of the $n$-th molecular eigenstate, coupled to this non-Hermitian potential is given by a Lorentzian function[62]

$$N_n(E) = \frac{\Gamma_n^2}{(E - \Lambda_n)^2 + \Gamma_n^2},\qquad(1)$$

where $E$ is the energy, and $\varepsilon_n = \Lambda_n + i\Gamma_n$ is the complex eigenenergy of the state[63]. The imaginary part $\Gamma_n$ represents the energy-broadening of the state and its lifetime is $\tau_n = \hbar/\Gamma_n$ (ref 62). Here we use this expression to estimate the lifetimes from the computed DOS. We implicitly consider the DFT Kohn–Sham eigenenergies obtained within the generalized gradient approximation to be a good approximation of the real quasi-particle energies. This is true in particular for states around the Fermi level of a metal[64] and for occupied molecular states[65], if the inherent DFT self-interaction error is corrected for (ref. 37). Instead, one must be more careful in extending this assumption to unoccupied states at energies significantly above $E_F$. In fact the DFT

Kohn–Sham eigenenergies usually underestimates the transport gap of molecules, with the empty LUMO placed too low in energy. If the DFT LUMO falls in an energy region where there are also metal $d$-bands, this may result in an artificially large broadening and spin-splitting, similarly to that found by Raman et al.[40]. Furthermore, for molecules near metallic surfaces the Kohn–Sham spectrum does not reproduce the transport gap reduction caused by image charge effects[48,49]. A fortuitous error cancellation between the LUMO underestimation and the lack of renormalization may sometimes result in a correct energy level alignment at a metal/molecule interface, however, this is not generally the case and corrective schemes are required. In this case, we use the scissor operator[50–52] in order to shift the unoccupied energy level to correct position which is estimated by c-DFT[50,51]. This is a well-established approach in transport and allows in practice to get the correct energy-level alignment between a molecule and metallic substrate. Finally, we also stress that within the DFT + NEGF, the lifetime is determined only by the molecule-surface coupling, while other effects, such as electron–electron interactions (beyond mean field), are neglected in our calculation of Γ. These effects can in principle be included in the description by introducing additional self-energies in the DFT-based transport formalism[65] and they would generally decrease the computed lifetime of a molecular state.

The DFT + NEGF Smeagol calculations were performed by using the Perdew, Burke, Ernzerhof generalized gradient approximation exchange-correlation density functional[66]. Standard double-zeta plus polarization basis sets together with Troullier-Martin pseudopotentials were employed. An equivalent real space mesh cutoff of 400 Ry and 2 by 2 **k**-points in the plane of the surface were considered.

The geometries of the interfaces considered in the DFT + NEGF calculations were previously optimized by using a DFT supercell approach. For this purpose, the Fritz Haber Institute ab initio molecular simulations (FHI-AIMS) all-electron code[67] was employed. The standard numerical atom-centered orbitals basis set 'tier 1' and 'tier 2' were considered for Co and H, C, N, O, Al, respectively. The fcc Co surface was built as a four-layer slab with a (4 × 4) square unit cell. Only the molecules and the first two Co layers were allowed to relax until forces were smaller than 0.01 eV/Ang. In order to include Van der Waals interactions (vdW), we employ the so-called DFT + vdWsurf[68] scheme, which extends the dispersion-corrected exchange-correlation density functional by Tkatchenko and Scheffler[69] (DFT + vdW) through the inclusion of the many-body collective response of the substrate[70]. Note that the basis sets used in FHI-AIMS are specifically designed for high accuracy in order to achieve total energy convergence of the all-electron system[67], and we have verified that the addition of counterpoise-corrections has a negligible effect on the computed binding energies. More details on the applications of the methods to Co surfaces can be found in a previous work[37]. The proper inclusion of vdW interactions is naturally crucial in order to describe physisorbed systems.

The c-DFT calculations have been carried out with a development version of the SIESTA code[60] and according to the procedure outlined in ref. 50 for systems with periodic boundary conditions. Norm-conserving Troullier-Martin pseudo-potentials together with basis sets of double-zeta plus polarization quality were employed. A 4 × 4 × 1 Monkhorst-Pack **k**-point mesh was used for the integration over the Brillouin zone. A constraint is applied to reduce the number of electrons in the metal by one, and at the same time increase it by one on the molecule. Such an approach is appropriate for the study of the second layer, since the physisorbed molecule is weakly coupled to the rest of the system, so that in the ground state it has the same integer number of electrons that it would have in the gas phase. Importantly, in 2PPE experiments the unoccupied $Alq_3$ states are effectively measured through the excitation of a single electron from the metal into the molecule, which is exactly the charge-transfer state calculated by c-DFT[50].

**Photoemission experiments.** Experiments were carried out on a Au(111)/Co/$Alq_3$ sample in a ultra-high vacuum (UHV) chamber with different methods of photoemission spectroscopy. The UHV chamber consists of one spectroscopy chamber with a base pressure of $4 \times 10^{-11}$ mbar and two evaporation chambers for Co and $Alq_3$ with a base pressure of $9 \times 10^{-10}$ mbar, respectively. The spectrometer chamber includes a cylindrical sector analyser (Focus CSA 300) for energy selection, with an energy resolution of 0.22 eV at 4 eV pass energy; the acceptance angle is ± 12°. The two evaporation chambers enable in situ preparation, resulting in clean surfaces and high-quality interfaces. An Au(111) single crystal was prepared by Ar-ion sputtering and consecutive annealing to 800 K. Forty MLs (3.5 nm) thick Co films were deposited afterwards by electron beam epitaxy and annealed to 600 K. This results in a hexagonal, bulk-like Co structure with an in-plane magnetic uniaxial anisotropy along the high-symmetry axis of Co. One nominal ML (one ML = 1.3 nm) $Alq_3$ thin films were deposited on the freshly prepared Co surface with a Knudsen cell. Deposition rates are monitored by a quartz crystal balance calibrated with ellipsometry. We employ UPS for the detection of occupied states, using the He I transition at 21.2 eV of a gas-discharge vacuum ultraviolet lamp (Focus HIS 13). For the detection of unoccupied states as well as for time-resolved measurements, we use 2PPE. As source we operate a Ti:Sa laser oscillator (Griffin 10, KM Labs) with a central wavelength of 780 nm (1.59 eV), delivering laser pulses of 35 fs with an energy of 10 nJ at a repetition rate of 80 MHz. Using a β—barium borate (BBO) crystal we create frequency-doubled light at 390 nm (3.2 eV) for the spectroscopic measurements. For time-resolved pump-probe measurements, the frequency-doubled light is send through a Mach–Zehnder interferometer, allowing to control the time delay between the arrival times of the probe pulse with respect to the pump pulse on the sample[41]. This way, the lifetime of hot electrons populating intermediate states during the 2PPE process can be extracted. The sample preparation and all measurements were performed at room temperature. In 2PPE a first photon (pump) is used to optically populate an excited state at the Co/$Alq_3$ spinterface. A subsequently absorbed second photon (probe) photoemits electrons from the transiently populated excited state. These photoemitted electrons can then be analysed with respect to their energy. By varying the delay between the pump and the probe photon, one can infer the femtosecond-to-picosecond transient dynamics[54].

**Rate equation model.** The population of the majority and minority spin channel ($N_u$ and $N_d$, respectively) of the uHIS can be modelled by this set of coupled differential equations:

$$\dot{N}_{u,d} = -\left[\frac{N_{u,d}}{\tau_{dep}} \pm \frac{N_u - N_d}{\tau_{sf}}\right]$$

$\tau_{dep}$ is the molecule-electrodes relaxation time, responsible for the decrease in the population of the uHIS (that is, the decrease of $N_u + N_d$), while $\tau_{sf}$ is the quasi-elastic spin-flip time. On the time scale $\tau_p = (1/\tau_{dep} + 2/\tau_{sf})^{-1}$ the population of the minority and majority carrier spins equilibrates. After solving these equations, the population of the majority and minority channels can be written as:

$$\frac{N_u}{N_0} = \frac{1}{2}\left(\exp^{-\frac{t}{\tau_{dep}}} + P_0 \exp^{-\frac{t}{\tau_p}}\right)$$

$$\frac{N_d}{N_0} = \frac{1}{2}\left(\exp^{-\frac{t}{\tau_{dep}}} - P_0 \exp^{-\frac{t}{\tau_p}}\right)$$

where $N_0$ is the initial population ($N_0 = (N_u + N_d)_{t=0}$) and $P_0$ is the initial spin polarization ($P_0 = ((N_u - N_d)/(N_u + N_d))_{t=0}$) of the uHIS.

**Data availability.** The data that support the findings of this study are available from the corresponding author on request.

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

## Acknowledgements

The experimental work carried out at the University of Kaiserslautern was partly funded by the SFB/TRR 173 *Spin + X*: spin in its collective environment (Project B05) from the DFG. A.D. and I.R. were sponsored by the European Union through the FP7 project 618082 ACMOL. S.S. acknowledges the European Research Council, Quest project, for financial support. P.T. and B.S. thankfully acknowledge financial support from the Graduate School of Excellence MAINZ (Excellence Initiative DFG/GSC 266). Computational resources were provided by the Trinity Centre for High Performance Computing (TCHPC) and the Irish Centre for High-End Computing (ICHEC).

## Author contributions

M.C., A.D., I.R., S. S. and M.A. conceived the project. A.D. and I.R. performed the DFT calculations. P.T., N.H., N.G., J.S. and B.S. performed the photoemission experiments. All authors analysed/discussed the results and wrote the manuscript.
