## [Peer Review File · Nature Communications]

Reviewers' comments:

Reviewer #1 (Remarks to the Author):

The authors present results of calculations for lifetimes of excited states in Alq₃@Co, which help to interpret experimentally observed data published before (Ref. 42). Their main conclusions are:
a) for the rationalization of the experimental data the weakly bound second layer of Alq₃ molecules is relevant
b) quasi-elastic spin-flip scattering is responsible for the (measured) lower decay rate of the minority spin.

Calculations concerning the electronic molecule/surface coupling are done with contemporary available state-of-the-art methods, the paper is well written and the conclusions in my opinion overall are reasonable. I think the topic and the results are of interest for at least parts of the readership of Nature Communications.

One should consider the following points:

1. Title: Does the paper really present "a new route for tuning spin-dependent transport in molecular spintronics"? - I think it mainly rationalizes previous experimental findings.

2. Central in the present work are those Alq₃ molecules that are bound to the others by weak (intermolecular) interactions. Concerning the description of these interactions and consequences for the intermolecular arrangements the work in my opinion is slightly below the present quantum-chemical standards. It is particularly surprising that - as noted by the authors on p.15 - including van-der-Waals interactions does not change the bond distance. For a molecule like Alq₃ I would have expected this type of interactions to be at least as important as interactions resulting from higher multipole moments. I would like to encourage the authors to use a larger basis set to make sure that the Alq₃ molecules in the calculations are held together by more than only the basis set superposition error. If possible, one could also perform a counterpoise-corrected optimization. In any case, the binding energies for molecules of the second layer should be given and discussed (and compared to kT). Further, it might be helpful to estimate the sensitivity of results to intermolecular distances by additionally calculating broadenings and resulting lifetimes for intermolecular distances in-/de-creased by several percent.

3. Terms like "strongly hybridized molecules" (p. 6) are very uncommon (wrong), at least in quantum chemistry. Maybe a quantum chemist should be consulted to check for further slips.

Reviewer #2 (Remarks to the Author):

This article presents a combined effort from the theoretical and the experimental side devoted to the understanding of the ferromagnetic-molecular interfaces for spin injection. In particular, the researchers focus on the Cobalt/Alq₃ interface, a suitable choice being this combination quite popular in recent years.

The work seems to be carefully done, but I think the authors need to make a more coherent effort for explaining the novelty of their findings in an independent manner. In particular, they should clarify what distinguishes this work from previous articles by Cinchetti et al.. Moreover, they compare these results with the important article by Raman, Moodera et al. While useful, such comparison is sometimes not well justified and the connection between both works seems tenuous at times.

Here below I present some specific issues which I believe they should be clarified/improved by the authors.

In the introduction of the article, image 1 is not extremely clear. What is the difference the author are aiming to target? What is actually the difference between the "metallic spin filter" and the "resistive one"? Would these not be the same in the case, for example, a second molecular layer is present in the

diagram 1a? I think the situation is quite more complex than such a simplistic diagram and probably most of the experimental cases lie somewhere in between both extremes.

The introduction contains some overstatements, for example: "very little effort (if nothing at all) has been devoted to the study of the mechanisms affecting hopping diffusion in solid state devices". I am not sure what the authors actually mean, as they disregard the intensive work performed in organic electronic devices in the last two decades. In my opinion, such bold statements do not help to set the work properly into context. Many of the references in the article are focus on organic spintronics, but the authors should look into a wider perspective and understand that organic electronics has produced valuable contributions along the years.

In any case, I can agree with the authors that a dynamic description of the ferromagnetic-organic interfaces is actually extremely important and as such, it has been studied in the past by the group of M. Cinchetti. In particular, his works of Nature Materials 8, 115 (2009) and Nature Physics 9, 243 (2013) are important milestones in the organic spintronics field. I would like to point out that in their Nature Physics article, the authors study the same system as in this current manuscript.

I am not an expert in electronic structure calculations, and as such I do not have a strong opinion about the quality of the methods employed in the article. However, I would be interested in understanding how the authors can cross-check the calculated geometries of the Alq3 molecules on the Co surface. I understand this point is quite challenging, but on the other hand the simulations rely on relaxed structures which have not been tested experimentally. Moreover, some parts of the text regarding the electronic structure calculations are slightly too didactic to my taste. The text becomes too long and some technical details could have been moved to the supplementary information.

One of the first conclusions of the article is that the first layer of molecules is mainly chemisorbed, while the second is physisorbed on top. I am afraid in this case I also struggle to understand the novelty on this point. Such conclusion has been put forward a number of years ago by several UPS/XPS studies of Alq3 molecules (see de Jong, for example, for organic spintronics) and it is now somewhat trivial.

The authors make a point regarding the importance of the spin lifetime in the second molecular layer, being this large value an important clue for spin injection. The point I do not have clear is what is the relation of this second molecular layer with the bulk (that is, with the subsequent molecular layers). This is a critical point for spin injection, as a device with just two molecular layers is likely to operate in the tunnelling regime.

Sometimes I believe the authors stress the term "dynamical" in an unreasonable way. Certainly, spin processes are dynamical and perhaps with indicating that in the introduction should be enough rather than pointing it out many times along the article. I agree that "dynamical" studies such as those performed by 2PPE are important in addition to "static" studies provided by other spectroscopic method, but that is simply a general statement. Moreover, the importance of the "dynamic" experiments congrats with the, I guess, "static" DFT calculations which I am not certain they can grasp the physics at the time scale. This particulate point writes me and should be carefully discussed and clarified.

As a final point, I am uncertain about the predictive power of the results presented here. The authors state "the exact determination of which microscopic mechanism leads to a more effective refilling of the uHIS is beyond the scope of this paper", and that would certainly be a nice result indeed. I keep wondering how much support and insight the authors are getting from the theory apart from some energy level determination and some complex geometry calculations which not seem to be that relevant anyway. In the last sentence, they write "the design of weakly-coupled organic layers..." but those design rules are absent in this current manuscript and would be really welcome by the scientific community working in this topic.

In general, I find the article valuable but I think the authors need provide a better claim of three results and polish multiple issues before it can be considered for publication in a demanding journal.

Reviewer #3 (Remarks to the Author):

This paper fundamentally elaborates upon an experimental paper (REF 42) recently published in

Nature Physics. The spin polarized photoemission in REF 42 was used to infer the existence of a hybrid interface state (HIS), but the lifetime of this state was observed to be rather long for the presumed strong coupling between molecule and substrate. In the present work, the relatively long lifetime measured in REF 42 for this state is explained by noting that the state seen actually arise in the second layer of molecules, which are less strongly coupled. This brings up the obvious question of whether the so-called uHIS is actually properly described as an interface state if it arises only in the second layer. I take away that the author's view is that the first layer is so strongly coupled as to be considered predominantly part of the Co electrode and that the only sensible definition of "hybrid" begin at the second layer. It took me several readings to get this idea though. I think this paper should be published in Nature Communications, but I would strongly encourage the authors to focus their presentation (specific recommendations below) and consider some other minor revisions. The fact that this long-lived spin polarized HIS arises in the second layer is a new and interesting development, perhaps connected to the Forrest group's conceptual idea that the rate limiting step in charge injection is often related to multiple layers near the interface (Baldo and Forrest Phys Rev B 65, 085201(2001)) instead of just the first layer. I feel that the current organization of the paper struggles to set out this fundamentally new idea for spintronics in a clear way. This applies to both the motivating discussion in the introduction and even to the title which does not quite emphasize the unique role of the second layer in spin dependent interface properties. In my opinion, it is not the "weak coupling" alone that makes this paper interesting but the specific realization of weak coupling within the second layer.

A specific recommendation would be to introduce the question of spin polarization in the near surface region beyond the first layer in the second paragraph on page 2 as a lead-in to the extended background on the ideas from Figure 1 taken from Raman et al. In addition, I think the title could be re-worked to emphasize this evidently unique and somewhat unexpected 2nd layer effect.

The new results presented seem to be computational with relatively minor support from new (spin averaged) UPS and 2PPE desorption observations. The paper would be vastly stronger if the PES desorption studies used the SPLEED detector from REF 42 to directly access spin dependent populations as the 2nd layer is desorbed and then regrown. Perhaps this is not possible for some reason? Can the authors please comment on this? This is the weakest point of the paper.

DFT-NEGF studies (with scissor corrections) show an enhanced broadening of unoccupied states in the strongly chemisorbed first layer compared to the more weakly adsorbed second layer (compare Figure 2 and Figure 3). I think this is the first DFT study I have seen that incorporates a second layer of molecules in proximity to a magnetic electrode. This is an important development and a difficult computational challenge.

This should be relevant to both spin injection into bulk organic films and also to TMR effects in even very thin organic films. Incidentally, it would be nice if the authors would consider how their results can be placed in the context of Alq3 TMR spin valve studies (Szulczewski et al., Appl Phys Lett 95,022506,(2009)) which show an unusual thickness dependence where the only significant change is a slight reduction of TMR upon the initial 2nm deposition (onto a CoFe electrode).

Minor Comments:

- 1) First sentence of paragraph 2 page 2 needs a "the" before "spinterface".
- 2) The claim on page 6 that Alq3 makes a disordered first layer is not supported by data or citations. Direct observation of disorder within the first Alq3 monolayer on a copper surface has been found in STM studies of Wang et al. Org. Electronics 12, 1920 (2011).

REVIEWERS' COMMENTS:

Reviewer #1 (Remarks to the Author):

The paper has been carefully revised and the points required for publication with nature communications are overall fulfilled.

Concerning the term 'hybridization': In quantum chemistry this typically is used in connection with atoms (sp^2 , sp^3 , etc.) In the present context 'admixture of orbitals' maybe is more familiar for quantum chemists, but as 'hybridization' is apparently commonly used in the respective community and as it is understood also by neighboured communities it can be kept.

Reviewer #2 (Remarks to the Author):

I think the authors have done a good effort for explaining the novelty of their findings, which is now clear to me especially when compared to previous results from some of the current authors. Now the role of the second weakly interacting layer is explained more openly and that highlights its importance against previous spin filtering mechanisms.

The new introduction is more pleasing to me and I think it would help to reach a more diverse audience, interested in this field but not necessarily specialists. The new Figure 1, which I thin is crucial for the understanding of the article, is now more clear.

The response to my comments and to those of the other referees seems to be compelling and certainly well developed. The work developed seems robust and now it is more clearly outlined.

As a comment, I strongly support the term "strongly hybridised molecules" which has a clear meaning in the physics community, if I agree nonetheless that is not so common in the quantum chemistry environments.

In general terms, I think the article has been massively improved and I am happy to support its publication in the current form.

Reviewer #3 (Remarks to the Author):

The authors have done a diligent job of responding to my comments and those of other reviewers.

I stand by my initial recommnedation that the paper should be published in Nature Communications. It reports good quality results that support fundamentally new ideas in the field of organic spintronics. Such studies are sorely needed to support complex and hard-to-interpret device characterization.

Reply to the reviewers' comments:

Reviewer #1

Comment 1. Title: Does the paper really present "a new route for tuning spin-dependent transport in molecular spintronics"? - I think it mainly rationalizes previous experimental findings.

Reply to comment 1. The paper indeed rationalizes previous experimental findings on the spin-dependent lifetime of Co/Alq₃ interface states. In doing so, it shows that organic molecules belonging to the second layer which are weakly electronically coupled to the Co substrate, can act as spin filters even if the hybrid interface states are not spin-split. In this case, the spin filtering is due to the spin-dependent electron dynamics at the interface. This influences the injection of spin-polarized carriers across the interface and their successive hopping diffusion into further molecular layers. This mechanism is different from all other the spin-filtering mechanisms discussed up to now in literature, which apply only to the tunneling regime.

→ To reflect this extremely important, and new, finding, we have changed the title to: "Dynamic spin-filtering at the Co/Alq₃ interface mediated by weakly coupled second layer Alq₃ molecules".

→ Moreover, also following the suggestions of Referee #2 and #3, we have modified the introduction of the manuscript, in order to underline the novelty of our work. (See the reply to Referees #2 and #3).

Comment 2. Central in the present work are those Alq₃ molecules that are bound to the others by weak (intermolecular) interactions. Concerning the description of these interactions and consequences for the intermolecular arrangements the work in my opinion is slightly below the present quantum-chemical standards. It is particularly surprising that - as noted by the authors on p.15 - including van-der-Waals interactions does not change the bond distance. For a molecule like Alq₃ I would have expected this type of interactions to be at least as important as interactions resulting from higher multipole moments. I would like to encourage the authors to use a larger basis set to make sure that the Alq₃ molecules in the calculations are held together by more than only the basis set superposition error. If possible, one could also perform a counterpoise-corrected optimization. In any case, the binding energies for molecules of the second layer should be given and discussed (and compared to kT). Further, it might be helpful to estimate the sensitivity of results to intermolecular distances by additionally calculating broadenings and resulting lifetimes for intermolecular distances in-/de-creased by several percent.

Reply to comment 2

The description of interatomic interactions for molecules on metallic surfaces is a complex task, which requires a method that can deal with both covalent and van der Waals interactions and that is able to include the important collective effects (screening) present in the substrate. High-level quantum chemistry methods or many-body methods such as the random-phase approximation (RPA) for the correlation energy can in principle be used for this purpose. However, in practice, they either perform well for only one sub-system (the

molecules or the substrate) or they are too demanding for applications that comprise hundreds of atoms, such as the Co/Alq₃ interface. Because of these limitations we have used the DFT+vdW^{surf} method [Phys. Rev. Lett. **102**, 073005 (2009), Phys. Rev. Lett. **108**, 146103 (2012)]. This has been developed by Tkatchenko and co-workers for the specific purpose of describing van der Waals interactions in hybrid organic-inorganic interfaces. It determines the van der Waals parameters from the DFT charge density instead of using empirical or previously fitted values, and it includes the screening inside the metal. Over the last three years the method has been extensively tested and benchmarked against experimental results and other competing approaches [see for example: Phys. Rev. B **86**, 245405 (2012), New J. Phys. **15**, 053046 (2013), Phys. Rev. B **87**, 165443 (2013), Phys. Rev. B **88**, 035421 (2013), Phys. Rev. Lett. **111**, 106103 (2013), J. Phys. Chem. C **117**, 3055 (2013), Acc. Chem. Res. **47**, 3369 (2014), Phys. Rev. Lett. **115**, 086101 (2015), Phys. Rev. B **93**, 035118 (2016)]. The conclusion common to all these works is that indeed DFT+vdW^{surf} provides an accurate description of absorption geometries of molecules on metallic surfaces, and it outperforms other vdW-inclusive DFT approaches. In addition, applications to complex systems have demonstrated its predictive potential [Nature Materials **11**, 834 (2012), Nature Commun. **4**, 2569 (2013)]. This therefore shows that the methods used are at the cutting edge of the present ab initio modelling standards for adsorption of molecules on metal surfaces.

Regarding the Referee's comment that "It is particularly surprising that - as noted by the authors on p.15 - including van-der-Waals interactions does not change the bond distance.", we note that in the Methods Section we explain that "the absorption distance of *chemisorbed molecules* is not affected by the inclusion of the vdW interactions, although the binding energy usually is. Instead a proper inclusion of vdW interactions is naturally crucial in order to describe the *physisorbed systems*". In other words, only for the molecules directly in contact to the metal surface, and therefore strongly bonded, the inclusion of vdW interactions does not significantly change the bond distances. For the second molecular layer, which is bonded to the first one only by vdW forces, the inclusion of vdW interactions in our simulations is crucial.

In the case of first-layer chemisorbed molecules, the binding distance is mainly determined by the covalent bond between the molecule and the substrate. This is well described already with standard DFT functionals such as PBE, while the vdW interactions contribute mainly to the energetics. Instead, in the case of physisorbed molecules, the vdW interactions are almost entirely responsible for the absorption process, and geometry optimization cannot ignore them. For a physisorbed molecule in the supercell displayed in Fig. 3 the binding energy is only of 0.162 eV, when calculated with PBE, while it increases to 1.083 eV when vdW forces are included through the vdW^{surf} functional. This energy cannot be straightforwardly compared to room temperature kT in order to understand the desorption process. Such study would require us to consider entropic and kinetic effects, and it is well beyond the scope of our work, which instead focusses on the magnetic and electronic properties of the interface. Nevertheless, we note that the computed binding energy is very close to typical values measured for many other similar-size physisorbed molecules [see, for example, Phys. Rev. Lett. **116**, 146101 (2016)] which desorb at similar temperatures, and it is of the same order of the computed one for another case of a two molecular layer system (J. Chem Phys. C **117**, 3055 (2013)]. In contrast, for first-layer chemisorbed molecules the binding energy can be of the order of ten or tens of eV per molecule.

Regarding the basis sets, for the relaxation we have employed the FHI-AIMS DFT code. FHI-

AIMS uses basis sets specifically designed to achieve total energy convergence of all-electron systems, which require very high accuracy. For the basis used in this work (tier-2), absolute energies have an accuracy of 0.01 eV/atom and energy differences have an estimated systematic error of only 0.1 meV/atom. A detailed discussion on the accuracy of FHI_AIMS can be found in the original article presenting the implementation [Blum *et al.* Comput. Phys. Commun. **180**, 2175 (2009)], as well as in many other articles on molecular systems that have employed FHI-AIMS. In the above mentioned reference to the code, the basis set superposition error (BSSE) has been thoroughly discussed and a specific example for the binding energy of two water molecules is presented. The conclusion is that “BSSE corrections are not critical for well converged numerical atomic orbital-based total energies and energy difference that can be achieved already at a moderate basis size (tier 2)”, i.e. for the basis size utilized in this work. Nevertheless, in order to rule out the BSSE for our large and complex systems, we have carried out the counterpoise-corrected calculations suggested by the Referee. This means that we have calculated separately the *total energy* for the substrate and physisorbed molecule that form two “fragments” of the whole Co/Aq₃ system, once by including only the basis function for the atoms of the fragment in question, and once by including additional so-called ghost orbitals (GO) replacing the other fragment (for these tests we have considered cluster geometries instead of supercells, since ghost atoms cannot be included in periodic calculations in the current version of the code). The results are reported in the Table below and indicate that indeed BSSE changes energies by only a few meV, which is a negligible amount when compared to the binding energy. This confirms that the basis set superposition error is negligible in our calculations, and that our binding geometries and energies are converged.

GGA-PBE functional	Energy (eV) –without GO	Energy (eV) –with GO
Molecule	-45522.5954	-45522.5984
Substrate	-4918819.6560	-4918819.688

PBE+vdWsurf functional	Energy (eV) –without GO	Energy (eV) –with GO
Molecule	-45523.6670	-45523.6640
Substrate	-4918824.3253	-4918824.3269

Finally, we have tested the sensitivity of the life-time results for the molecular states with respect to the molecule-substrate distance, as suggested by the Referee. We have shifted the 2nd layer molecules away from the substrate and closer to the substrate by 0.5 Å, so that we effectively check variations of 10% of the molecule-substrate distance. Upon increasing the molecule-substrate distance by 0.5 Å, the broadening of the different peaks is in the range of 2-4 meV, while upon decrease of the distance by 0.5 Å it is in the range of 3-12 meV. Although, as expected, by stretching (shrinking) the molecule-substrate distance the broadening slightly decreases (increases), the test shows that the results are quite robust. In fact, the values of the broadening stay quite close to range indicated in the manuscript (3 to 6 meV).

→ We have added the following sentence in the methods section:

“Note that the basis sets used in FHI-AIMS are specifically designed for high accuracy in order to achieve total energy convergence of the all-electron system [Blum *et al.* Comput. Phys. Commun. **180**, 2175 (2009)], and we have verified that the addition of counterpoise-corrections has a negligible effect on the computed binding energies.”

→ We have added the following sentence in the main text:

“The binding energy per atom for the second layer molecule is found to be 30 meV, while for the molecules in the first layer it is an order of magnitude larger, which indicates that the second layer molecules will desorb significantly easier than the ones of the first layer.”

→ We have added the following sentence in the main text:

“To verify the robustness of the results we have repeated the calculation by increasing (decreasing) the 2nd layer molecule-substrate distance by 0.5 Å, and we have found that the changes in the computed broadening are within a factor two of the ones calculated at the relaxed distance, so that overall the predicted order of magnitude of the lifetime does not change. This demonstrates that our results are robust against systematic errors deriving from the computational approach employed for the structural relaxations.”

→ In the Methods section we have removed the sentence: “Notably, after several extensive tests, we found that the absorption distance of chemisorbed molecules is not affected by the inclusion of vdW interactions, although the binding energy usually is.” This indeed caused misunderstanding. Now we just state that “the proper inclusion of vdW interactions is naturally crucial in order to describe physisorbed systems.”

Comment 3. Terms like "strongly hybridized molecules" (p. 6) are very uncommon (wrong), at least in quantum chemistry. Maybe a quantum chemist should be consulted to check for further slips.

Reply to comment 3.

The term “strongly hybridized” is commonly used in both theoretical and experimental surface science literature for organic electronics/spintronics, exactly with the same meaning used here. In the Methods section we provide a mathematical definition for the hybridization strength and lifetime that dates back to a seminal work of Anderson [Phys. Rev. 124, 41 (1961)] and Newns (Ref. 62). The hybridization strength can be mathematically characterized through the substrate self-energy that indeed is often referred as “hybridization function” in the physics literature [see for example Phys. Rev. B **85**, 085114 (2012)], with no possible misunderstanding. However, if the Referee still feels that this definition can be misleading in specific scientific communities, we would be thankful if he/she could suggest an alternative terminology and we will be pleased to consider that.

Reviewer #2

Comment 1. The work seems to be carefully done, but I think the authors need to make a more coherent effort for explaining the novelty of their findings in an independent manner. In particular, they should clarify what distinguishes this work from previous articles by Cinchetti et al. Moreover, they compare these results with the important article by Raman, Moodera et al. While useful, such comparison is sometimes not well justified and the connection between both works seems tenuous at times.

Reply to Comment 1. We would like to thank the referee for pointing out that the main message of the manuscript was not stated in a clear way.

→ We have taken this comment extremely seriously and have completely re-written the introduction. In the new version, we explain the novelty of our finding both with respect to previous articles by Cinchetti et al, as well as with respect to the work of other groups (Moodera, Raman etc.).

In brief, literature up to now has only discussed the spin-filtering properties of spinterfaces in the tunneling regime (the so-called metallic and resistive spin-filtering in Ref. 11). Starting from the experimental observation of a spin-dependent lifetime for the Co/Alq₃ interface (Ref 41), we have studied now the origin of such lifetimes and their implications for molecular spintronics devices. We find that the spin-dependent lifetime of the Co/Alq₃ interface originates from hybrid interface states that are localized on the weakly coupled second molecular layer. Such states are not spin-split (in contrast to the states described by Moodera and Atodiresei), but acquire a spin-dependent lifetime due to dynamical spin-relaxation driven by the interaction with the cobalt substrate. This effect constitutes a new spin-filtering mechanism, which will dominate when electrons are injected at high voltages into hybrid interface states and diffuse by hopping into further molecular layers of molecular spintronics devices. As this injection/hopping regime has never discussed before, we believe that our manuscript provides a significant advancement in the field of molecular spintronics.

Comment 2. In the introduction of the article, image 1 is not extremely clear. What is the difference the authors are aiming to target? What is actually the difference between the "metallic spin filter" and the "resistive one"? Would these not be the same in the case, for example, a second molecular layer is present in the diagram 1a? I think the situation is quite more complex than such a simplistic diagram and probably most of the experimental cases lie somewhere in between both extremes.

Reply to Comment 2. We are sorry that figure 1 was misleading. Clearly, the division between "metallic" and "resistive" spin filtering is only a simplification that was used in literature (see e.g. Ref. 11) to describe how spinterfaces can possibly influence spin-dependent tunneling of electrons in devices. Our manuscript introduces a third limiting case, that we have called "dynamic" spin filtering. This new scenario will play the dominant role in the injection/hopping regime.

→ We have prepared a new version of figure 1, where the concept of "metallic", "resistive" and "dynamic" spin filtering is presented in a clearer way. Moreover, we have also rewritten the entire introduction, in order to explain these concepts and how our findings represent a conceptually new spin-filtering mechanism at hybrid interfaces formed with ferromagnetic metals.

Comment 3. The introduction contains some overstatements, for example: "very little effort (if nothing at all) has been devoted to the study of the mechanisms affecting hopping diffusion in solid state devices". I am not sure what the authors actually mean, as they disregard the intensive work performed in organic electronic devices in the last two decades. In my opinion, such bold statements do not help to set the work properly into context. Many of the references in the article are focus on organic spintronics, but the authors should look into a wider perspective and understand that organic electronics has produced valuable

contributions along the years.

In any case, I can agree with the authors that a dynamic description of the ferromagnetic-organic interfaces is actually extremely important and as such, it has been studied in the past by the group of M. Cinchetti. In particular, his works of *Nature Materials* **8**, 115 (2009) and *Nature Physics* **9**, 243 (2013) are important milestones in the organic spintronics field. I would like to point out that in their *Nature Physics* article, the authors study the same system as in this current manuscript.

Reply to Comment 3. We agree that the introduction of the article was not written in a clear way and that references to the important advances in the field of organic electronics were missing.

→ As already mentioned in the reply to comment 2, we have now rewritten the introduction, also considering the important points concerning organic electronics raised by the referee.

Comment 4. I am not an expert in electronic structure calculations, and as such I do not have a strong opinion about the quality of the methods employed in the article. However, I would be interested in understanding how the authors can cross-check the calculated geometries of the Alq3 molecules on the Co surface. I understand this point is quite challenging, but on the other hand the simulations rely on relaxed structures which have not been tested experimentally. Moreover, some parts of the text regarding the electronic structure calculations are slightly too didactic to my taste. The text becomes too long and some technical details could have been moved to the supplementary information.

Reply to Comment 4. In recent years, there has been an enormous progress in the theoretical modeling by DFT of molecules absorbed on surfaces. This has demonstrated that state-of-the-art methods can indeed achieve accurate results when compared with accurate experimental benchmarks [see, for example, the following references that investigate specifically the computational method used in the present work: *Phys. Rev. B* **86**, 245405 (2012), *New J. Phys.* **15**, 053046 (2013), *Phys. Rev. B* **87**, 165443 (2013), *Phys. Rev. B* **88**, 035421 (2013), *Phys. Rev. Lett.* **111**, 106103 (2013), *J. Phys. Chem. C* **117**, 3055 (2013), *Acc. Chem. Res.* **47**, 3369 (2014), *Phys. Rev. Lett.* **115**, 086101 (2015), *Phys. Rev. B* **93**, 035118 (2016)]. This has generated confidence about the predictive capabilities of state-of-the-art DFT methods, so that one can indeed study novel systems. We note however that we did cross-check a number of our results with experimental data. In fact, to begin with, one can investigate whether the simulated interfaces geometries are consistent at all with the available spectroscopic (e.g. XPS) data. This has been discussed in our previous work (Ref. 37), as well as in an article by other authors [*Adv. Mater.* **22**, 1626 (2010)] showing that, at least for the first layer, the geometry obtained by DFT is in agreement with the molecule-substrate binding that emerges by the analysis of the XPS spectra. Then, we must note that the robustness of the theoretical results for the electronic structure can be double-checked against the uncertainties in the molecular geometries. For example, we have tried to vary the molecule-surface distance by ± 0.5 Å, and found that the predicted lifetimes for the second layer stay of the same order of magnitude. See also the reply to the second comment of Reviewer 1.

We thank the referee for his/her comment about our manuscript being too didactic in the theoretical part. We have taken the suggestion on board and moved a number of paragraphs about the computational details to the method section.

→ The paragraph “The relative energy shift and broadening of the LUMOs depend on the strength of the Co- \mathbf{N}_1 , \mathbf{N}_2 and \mathbf{N}_3 coupling. [...] The energy splitting between these two peaks is largely affected by the distortion of the ligands, which determines a shift of the LUMO localized over \mathbf{N}_2 towards energies higher than that of the LUMO localized over \mathbf{N}_3 .” has been moved from the main text to the caption of Fig. 2.

The entire paragraph “Before moving to the experimental verification of these findings, [...] and then the DFT KS eigenvalues are shifted to the correct energy-positions through the application of a scissor operator (see Methods section for details).” has been moved to the Methods section. In the main text, we now just mention that the calculations have been done by using the scissor operator.

Only the results of the lifetimes computed with the scissor operator are now presented in the main text. This means that the left panel in Fig. 3 has been removed.

Comment 5. One of the first conclusions of the article is that the first layer of molecules is mainly chemisorbed, while the second one is physisorbed on top. I am afraid in this case I also struggle to understand the novelty on this point. Such conclusion has been put forward a number of years ago by several UPS/XPS studies of Alq3 molecules (see de Jong, for example, for organic spintronics) and it is now somewhat trivial.

The authors make a point regarding the importance of the spin lifetime in the second molecular layer, being this large value an important clue for spin injection. The point I do not have clear is what is the relation of this second molecular layer with the bulk (that is, with the subsequent molecular layers). This is a critical point for spin injection, as a device with just two molecular layers is likely to operate in the tunneling regime.

Sometimes I believe the authors stress the term "dynamical" in an unreasonable way. Certainly, spin processes are dynamical and perhaps with indicating that in the introduction should be enough rather than pointing it out many times along the article. I agree that "dynamical" studies such as those performed by 2PPE are important in addition to "static" studies provided by other spectroscopic method, but that is simply a general statement. Moreover, the importance of the "dynamic" experiments congrats with the, I guess, "static" DFT calculations which I am not certain they can grasp the physics at the time scale. This particulate point writes me and should be carefully discussed and clarified.

Reply to Comment 5. We believe that the lack of clarity about the role of the second layer and about the meaning of “dynamical” spin-filtering was largely due to the previous version of the introduction. As explained above, the new introduction makes now clear that in our manuscript we are not considering the tunneling regime but the injection/hopping regime. Here, the hybrid interface states are populated by electrons, in the same way as they are populated in the 2PPE experiments. In our combined experimental and theoretical effort, we demonstrate that the spin-dependent population of hybrid interface states of second layer molecules is determined by spin-flip processes mediated by the cobalt substrate (as depicted in Fig. 7).

Regarding the calculations, we note that the DFT+NEGF method is specifically designed to

describe electron flow in and out of the system at any time scale. It captures this flow of electrons from the molecule to the bulk of the electrodes via the addition of so-called electrode self-energies [see Phys. Rev. B **73**, 085414 (2006)]. In the Methods section we provide a mathematical definition for the methods used to evaluate the hybridization strength between the molecule and the substrate, and for the lifetime, that dates back to a seminal work of Anderson [Phys. Rev. **124**, 41 (1961)] and Newns (Ref. 62).

→ Besides changing the introduction, we have clarified the meaning of “dynamic” spin filtering in the discussion of our results on page 13.

Comment 6. As a final point, I am uncertain about the predictive power of the results presented here. The authors state "the exact determination of which microscopic mechanism leads to a more effective refilling of the uHIS is beyond the scope of this paper", and that would certainly be a nice result indeed. I keep wondering how much support and insight the authors are getting from the theory apart from some energy level determination and some complex geometry calculations which not seem to be that relevant anyway. In the last sentence, they write "the design of weakly-coupled organic layers..." but those design rules are absent in this current manuscript and would be really welcome by the scientific community working in this topic.

Reply to Comment 6. The theoretical calculations demonstrate that the lifetime of the 2nd layer molecules corresponds to the measured lifetimes, while the lifetime of the first layer molecules is orders of magnitude lower. This is a key aspect of our results. In fact, it was such result that guided us to investigate experimentally the details of the 2nd molecular layer. We are quite certain that the referee will appreciate the predictive power of our results after reading the revised version of the manuscript. In fact, we now point out much more clearly that we have found an effect that will dominate the spin-dependent injection of electrons in molecular spintronics devices. We have shown that in this scenario, spin-filtering is determined by second layer (physisorbed) organic molecules that acquire a spin-dependent lifetime due to spin-flip processes mediated by the substrate. In the conclusion, we propose how to influence such spin-dependent lifetimes, for example by introducing molecules with a high magnetic moment as second layer molecules. Up to now the role of second layer molecules has been mostly neglected, and all the efforts have been concentrated on adsorbing e.g. single molecule magnets directly on the ferromagnetic surface. However, physisorbed second layer molecules constitute a very promising playground for the design of advanced molecular spintronics concepts since:

- (i) they mostly keep their molecular character, contrary to the case of chemisorbed molecules;
- (ii) their physical properties, and thus the related interface spin filtering properties, can be simply tuned by acting on the molecules using external stimuli (such as light, doping, electric or magnetic fields)

→ We have added a paragraph in the conclusions (pages 13 and 14), where we discuss more in detail the potential of second-layer molecules for molecular spintronic devices.

Reviewer #3

Comment 1. The fact that this long-lived spin polarized HIS arises in the second layer is a new and interesting development, perhaps connected to the Forrest group's conceptual idea that the rate limiting step in charge injection is often related to multiple layers near the interface (Baldo and Forrest Phys Rev B 65, 085201(2001)) instead of just the first layer. I feel that the current organization of the paper struggles to set out this fundamentally new idea for spintronics in a clear way. This applies to both the motivating discussion in the introduction and even to the title which does not quite emphasize the unique role of the second layer in spin dependent interface properties. In my opinion, it is not the "weak coupling" alone that makes this paper interesting but the specific realization of weak coupling within the second layer.

A specific recommendation would be to introduce the question of spin polarization in the near surface region beyond the first layer in the second paragraph on page 2 as a lead-in to the extended background on the ideas from Figure 1 taken from Raman et al. In addition, I think the title could be re-worked to emphasize this evidently unique and somewhat unexpected 2nd layer effect.

Reply to Comment 1. We would like to thank the referee for pointing out this extremely important point, and for suggesting us a way to emphasize the relevance of our results.

→ Following the Referee's suggestions, we have completely rewritten the introduction and changed Figure 1 of the manuscript. In the new version, we now emphasize that weakly coupled, second layer organic molecules can act as spin filters even if the hybrid interface states that they form at the interface are not spin-split. The spin filtering is, in this case, due to the spin-dependent electron dynamics at the interface, and will influence the injection of spin-polarized carriers across the interface and their successive hopping diffusion into further molecular layers. This mechanism is different from all the other spin-filtering mechanisms discussed up to now in the literature, which apply only for the tunneling regime. In this sense, the dynamic spin filtering mechanism that we describe is the spin-dependent extension of the charge injection model of Baldo and Forrest, as correctly pointed out by the Referee. This important message is also captured by the new version of Figure 1.

→ According to the considerations above, and following the suggestion of the Referee, we have also changed the title of the manuscript into "Dynamic spin-filtering at the Co/Alq₃ interface mediated by weakly coupled second layer Alq₃ molecules".

→ To emphasize the unique role of the second layer molecules for spin injection at the Co/Alq₃ interface, we have also modified Fig. 7. The new version illustrates the dynamic process leading to spin-filtering in a clearer way.

Comment 2. The new results presented seem to be computational with relatively minor support from new (spin averaged) UPS and 2PPE desorption observations. The paper would be vastly stronger if the PES desorption studies used the SPLEED detector from REF 42 to directly access spin dependent populations as the 2nd layer is desorbed and then regrown. Perhaps this is not possible for some reason? Can the authors please comment on this? This

is the weakest point of the paper.

Reply to Comment 2. The main purpose of the experimental data presented in the manuscript is to demonstrate experimentally the theoretical prediction that the long spin-dependent lifetimes reported in Ref. 41 originate from weakly coupled, second layer Alq₃ molecules, and not from strongly chemisorbed, first layer Alq₃ molecules. This fact is clearly demonstrated by the spin-integrated time-resolved 2PPE measurements shown in Figure 6c. Here we clearly show that the “wings” associated to the long spin-dependent lifetimes are only present in the non-annealed Co/Alq₃ sample, while the signal from the sample annealed at 245°C, i.e. where the second layer molecules have been desorbed, is virtually the same as the signal from the bare cobalt substrate.

Adding spin-resolution to this data would not significantly change this scenario. In fact, the first layer of Alq₃ molecules is so strongly chemisorbed that it behaves as a metallic layer. Due to the extremely short lifetimes of electrons in the metal layer (only some femtoseconds) it would be almost impossible to distinguish between the spin-dependent lifetime of the electrons in the Co/Alq₃ interface (formed with 1st layer molecules) and the spin-dependent lifetime of electrons in the cobalt layers below (see also *Ultrafast spin-dependent electron dynamics in fcc Co*, Phys. Rev. Lett. 79 (1997) 5158). These two signals are superimposed onto each other and, since the lifetimes are comparable, it is impossible to disentangle these two channels.

In other words, the new measurements would not provide any additional information, due to the impossibility to disentangle the lifetimes from the cobalt and from the Co/1st layer (strongly chemisorbed) Alq₃ interface. We strongly believe that the revised version contains all necessary information to demonstrate the new dynamical spin-filtering mechanism mediated by second layer molecules at the Co/Alq₃ interface.

Comment 3. DFT-NEGF studies (with scissor corrections) show an enhanced broadening of unoccupied states in the strongly chemisorbed first layer compared to the more weakly adsorbed second layer (compare Figure 2 and Figure 3). I think this is the first DFT study I have seen that incorporates a second layer of molecules in proximity to a magnetic electrode. This is an important development and a difficult computational challenge.

This should be relevant to both spin injection into bulk organic films and also to TMR effects in even very thin organic films. Incidentally, it would be nice if the authors would consider how their results can be placed in the context of Alq₃ TMR spin valve studies (Szulcowski et al., Appl. Phys. Lett. 95, 202506, (2009)) which show an unusual thickness dependence where the only significant change is a slight reduction of TMR upon the initial 2nm deposition (onto a CoFe electrode).

Reply to comment 3. We are glad and thankful that the Referee appreciates our novel theoretical developments, which indeed allowed us to tackle this difficult computational challenge.

We agree with the Referee that in principle second layer molecules will affect not only injection and transport but also tunneling through the interface. This is mostly the case when the molecular orbitals of the second layer have a finite energetic spin splitting, as discussed in the literature (Ref. 40 of the manuscript). In our manuscript we concentrate on a lifetime effect that plays a role even when the molecular orbitals are not spin-split, but have a different spin-dependent lifetime. This effect will mostly affect spin injection and spin transport.

→ These concepts have been clarified in the new introduction of the paper, together with a new version of Fig. 1 and Fig. 7.

Regarding the paper of Szulczewski *et al.*, it is indeed interesting that the measured (T)MR ratio in their devices changes only upon initial deposition of 2nm Alq₃ and it is more or less constant for thicker organic layers. In our opinion, the experiment demonstrates two aspects. Firstly, spin-filtering in organics is an interfacial effect and indeed it is different in the tunneling regime (organic thickness <2 nm) and when one has real spin-injection into the organic. Secondly, the spin-relaxation length is longer than the thickness of the considered organic layer (2, 4 and 8 nm). This is consistent with the 30 nm spin-relaxation length estimated for Alq₃ (Nature Materials **8**, 693). Yet, unfortunately, from the mentioned experiment, it is not possible to extract any information about the actual spin-filtering mechanism. Furthermore, we note that the larger measured magneto-resistance is for quite small voltages (about 0.2 mV) indicating that the transport may happen through an impurity band filling the HOMO-LUMO gap and pinned at the Fermi energy, while here we discuss spin-injection into the LUMO, which happens at much larger voltages (of the order of 1 V).

→ The paper of Szulczewski et al. is a good example of spintronic devices working in the injection regime. We have added a reference to this paper in the introduction of the manuscript, where injection-based spintronics devices are discussed.

Comment 4. First sentence of paragraph 2 page 2 needs a "the" before "spinterface".

Reply to comment 4. Thank you.

Comment 5. The claim on page 6 that Alq₃ makes a disordered first layer is not supported by data or citations. Direct observation of disorder within the first Alq₃ monolayer on a copper surface has been found in STM studies of Wang et al. Org. Electronics 12, 1920 (2011).

→ We have added the reference suggested by the Referee.